# Predicting Progression of Kidney Injury Based on Elastography Ultrasound and Radiomics Signatures

**DOI:** 10.3390/diagnostics12112678

**Published:** 2022-11-03

**Authors:** Minyan Zhu, Lumin Tang, Wenqi Yang, Yao Xu, Xiajing Che, Yin Zhou, Xinghua Shao, Wenyan Zhou, Minfang Zhang, Guanghan Li, Min Zheng, Qin Wang, Hongli Li, Shan Mou

**Affiliations:** 1Molecular Cell Laboratory for Kidney Disease, Department of Nephrology, Shanghai Peritoneal Dialysis Research Center, Uremia Diagnosis and Treatment Center, Renji Hospital, School of Medicine, Shanghai Jiao Tong University, Shanghai 200127, China; 2School of Medicine, Department of Ultrasound, Renji Hospital, Shanghai Jiao Tong University, Shanghai 200127, China; 3China-Japan Friendship Hospital, Department of Ultrasound, Beijing 100029, China

**Keywords:** tissue elasticity imaging, kidney prognosis, regression analysis, machine learning

## Abstract

Background: Shear wave elastography ultrasound (SWE) is an emerging non-invasive candidate for assessing kidney stiffness. However, its prognostic value regarding kidney injury is unclear. Methods: A prospective cohort was created from kidney biopsy patients in our hospital from May 2019 to June 2020. The primary outcome was the initiation of renal replacement therapy or death, while the secondary outcome was eGFR < 60 mL/min/1.73 m^2^. Ultrasound, biochemical, and biopsy examinations were performed on the same day. Radiomics signatures were extracted from the SWE images. Results: In total, 187 patients were included and followed up for 24.57 ± 5.52 months. The median SWE value of the left kidney cortex (L_C_median) is an independent risk factor for kidney prognosis for stage 3 or over (HR 0.890 (0.796–0.994), *p* < 0.05). The inclusion of 9 out of 2511 extracted radiomics signatures improved the prognostic performance of the Cox regression models containing the SWE and the traditional index (chi-square test, *p* < 0.001). The traditional Cox regression model had a c-index of 0.9051 (0.8460–0.9196), which was no worse than the machine learning models, Support Vector Machine (SVM), SurvivalTree, Random survival forest (RSF), Coxboost, and Deepsurv. Conclusions: SWE can predict kidney injury progression with an improved performance by radiomics and Cox regression modeling.

## 1. Introduction

Chronic kidney disease (CKD) has become a global health burden, with an incidence of around 10% [1]. Progression to CKD at over stage 3 was estimated to cost USD 5367 to USD 53,186 per patient per year, a 1.3 to 2.4 fold increase compared with CKD stages 1–2, whereas the costs associated with end-stage renal disease were the highest, ranging from USD 20,110 to USD 100,593 [2]. However, the rate of CKD progression differs individually. Acute kidney injury is one of the major causes of and accelerating factors in CKD [3]. Thus, the determination of early predictors of the progression of kidney injury is important. However, current monitoring methods for the progression of kidney disease are not ideal. These include biopsy (which is too invasive to repeat), eGFR (which is only elevated after most kidney cells lose regenerative capacity and is insensitive to CKD progression), and proteinuria (which is largely affected by etiology and insensitive and non-specific to CKD progression) [3,4,5]. Studies on imaging techniques and urinary biomarkers are emerging as part of the search for promising non-invasive monitoring methods.

Ultrasound remains the preferred non-invasive radiographic method for diagnosing CKD due to its economic and portable properties [6]. Two-dimensional shear wave elastography (SWE) is an emerging technique of elastography ultrasound for evaluating kidney stiffness [7,8]. Based on the physical theory that shear wave propagation velocity is higher in stiffer tissues, the stiffness of kidneys can be estimated by the linear formula of Young’s modulus using shear wave velocity obtained through SWE [9,10]. Due to the anisotropy of the kidneys, the SWE parameters usually include Young’s modulus value in the cortex and Young’s modulus in the medulla [11]. However, a contradictory relationship between SWE and eGFR or histological fibrosis was found among different cohorts with different confounding factors [12]. Furthermore, few studies have reported the predictive value of SWE for the prognosis of CKD.

Recent advances in the radiomics analysis of ultrasound images of fibrosis [13,14] and artificial intelligence in clinical diagnostic and prognostic models [15] may provide a new approach to clarifying the relationship between SWE and CKD, as well as alternative CKD progression predictors to kidney biopsy. PyRadiomics is an open-source platform based on Python that has been widely used in radiology, including images of CT and MRI, and shows no difference from ultrasomics in ultrasonography [16,17]. PyRadiomics can extract high-throughput quantitative features from the region of interest (ROI) in medical images, including ultrasound images [16,17,18]. The extracted signatures by PyRadiomics include classes of first-order statistics (19 features), shape descriptors (including 2D and 3D, not often used in ultrasound), and texture classes of gray level cooccurrence matrix (glcm, 24 features), gray level run length matrix (glrlm, 16 features), gray level size zone matrix (glszm, 16 features), gray level dependence Matrix (gldm, 14 features), and neighboring gray tone difference Matrix (ngtdm, 5 features), based on original images or preprocessed images using built-in filters. Support vector machine (SVM), SurvivalTree, and Random Survival Forest (RSF) are machine learning models developed for clinical survival analysis based on binary classification [19,20,21]. Coxboost and DeepSurv are machine learning or deep learning models developed based on the traditional Cox regression method [22,23].

Hence, we observed the predictive value of SWE for CKD progression in our kidney biopsy cohort. We used the clinical index and pathological changes as references. We also applied a PyRadiomics analysis of SWE ultrasound images, traditional Cox regression models, and machine learning models in our study. We hypothesized that SWE would predict CKD progression with or without the help of radiomics and machine learning.

## 2. Materials and Methods

### 2.1. Study Design and Population

The study featured a prospective cohort of kidney biopsy patients in our hospital from May 2019 to June 2020. The inclusion criteria were patients aged 18–70 with unexplained abnormal kidney function, proteinuria over 1 g/day, rapidly progressive glomerulonephritis, or persistent hematuria with proteinuria. The exclusion criteria were patients who could not cooperate with breath-holding for SWE, contraindication for kidney biopsy (solitary or horseshoe kidney, bilateral kidney atrophy, bleeding tendency, severe hypertension, or acute pyelonephritis), pregnancy, comorbidities of cysts, urological stones or tumors, unilateral kidney atrophy, or pathological diagnosis of acute kidney injury. B-mode, SWE, and collection of serum and urinary samples from patients were performed on the day of the kidney biopsy. The included patients were then re-examined every three months for the first year and every six months thereafter. The primary outcome was the initiation of renal replacement therapy or death. The secondary outcome was CKD stage over 3. CKD stage was evaluated based on eGFR (MDRD) as CKD 1, ≥90; CKD 2, 60–89; CKD 3, 30–59; CKD 4, 15–29; and CKD 5, <15 at the time of inclusion and follow-up [24]. CKD progression was defined, according to the 2012 KDIGO Guideline [25], as a sustained decrease (measured at least twice, with >3 months in between) of eGFR over 25% from baseline accompanied by a drop in the CKD stage. The follow-up time was until 31 March 2022. Details of the study flow are in Appendix A.

### 2.2. Clinical, Pathological, and Ultrasound Index

Serum creatinine (Scr) and urinary creatinine (UCr) were measured by the sarcosine oxidase method. Urinary albumin was measured by the immunoturbidimetric method. The urinary albumin–creatinine ratio (ACR) was subsequently calculated. The eGFR was calculated using the MDRD equation [26].

B-mode and SWE ultrasound examinations were conducted by one ultrasound radiologist who had received specialized training in kidney ultrasound and SWE for more than five years. The radiologist was blind to patients’ clinical information. B-mode was performed before SWE as a reference for kidney imaging [27]. SWE was then carried out with patients in a right lateral decubitus position by Supersonic Imagine Aixplorer (convex transducer SC6-1, frequency 1–6 mhz). All patients were required to perform deep inspiration and breath-holding during the examination to ensure stable image generation and acquisition. The penetration depth of the radiofrequency data was 12 cm without the effect of BMI (Appendix A). In total, four images were acquired at the inferior pole of the left kidney of each patient. Regions of interest (ROI) were manually drawn at the kidney cortex, medulla, and sinus with fixed diameters of 6, 6, and 4 mm, respectively. Mean and Median SWE values of the kidney cortex, medulla, and sinus were calculated.

A kidney biopsy was performed under the guidance of B-mode ultrasound at the inferior pole of the left kidney. Biopsy specimens were formalin-fixed before routine hematoxylin and eosin, periodic acid–Schiff, periodic Schiff–methenamine silver, and immunofluorescence staining, or fixed with 4% PFA and 2.5% glutaraldehyde before observation under electron microscopy. Pathological changes were divided into 13 categories, scored, and diagnosed by two renal pathologists with over 20 years of experience. Grades of chronic changes were then determined according to total renal chronicity score as minimal, 0–1; mild, 2–4; moderate, 5–7; and severe, ≥8, by referring [28,29].

### 2.3. Radiomics Signature Extraction from Ultrasound

Considering the heterogeneity of kidney histological changes during CKD and the purpose of adding values to SWE, ROIs for radiomics analysis were manually drawn using 3DSlicer based on the ROIs of the SWE ultrasound images (Figure 1). Radiomics signatures of each ROI were then extracted by PyRadiomics (v3.0.1) [16,17]. Eight hundred and thirty-seven radiomics signatures were acquired from each ROI, including first-order statistics (18 signatures), gray level cooccurrence matrix (glcm, 24 signatures), gray level dependence matrix (gldm, 14 signatures), gray level run length matrix (glrlm, 16 signatures), gray level size zone matrix (glszm, 16 signatures), and neighboring gray-tone difference matrix (ngtdm, 5 signatures) calculated in original or wavelet-transformed (HHH, HHL, HLH, HLL, LHH, LHL, LLH, LLL) images. Median values of the radiomics signatures at the kidney cortex, medulla, and sinus from the 4 SWE images were used for final analysis.

### 2.4. Cox Regression, Machine Learning, and Deep Learning Modeling

Lasso regression and Cox regression were conducted using glmnt (v4.1–3) and survival (v3.2–10) in R (v4.0.5) and SPSS (v26). Features with likelihood test *p* < 0.1 in univariate Cox regression entered further multivariate Cox regression using the stepwise-backward method. The hypothesis of proportional hazard was verified by the chi-square test before Cox regression modeling. The likelihood test was also used in the comparison of Cox regression models. Nomogram was built based on the results of multivariate Cox regression. Machine learning models SVM, RSF, SurvivalTree, Coxboost, and deep learning model DeepSurv were built using the same features as the Cox regression model by the scikit-survival package (v0.17.1) of Python (v3.7.0). The dataset was split randomly at an 8:2 ratio, 80% for training and 20% for testing to prevent overfitting of the machine learning or deep learning models. Hyperparameters were optimized through grid search and 10-fold cross-validation. Models were evaluated by the concordance index (C-index). C-index >0.9 means a model with high accuracy; 0.7–0.9 means medium accuracy; 0.5–0.7 means poor accuracy. Ninety-five percent confidence intervals were calculated by Bootstrap 1000 times.

### 2.5. Statistical Analysis

Statistical analysis was performed by SPSS (v26) and R (v4.1.0). Normality distribution was tested by the Kolmogorov–Smirnov method before the Students’ *t*-test for normally distributed and Mann–Whitney U test for non-normally distributed data. The chi-square test was used for the comparison of categorical variables. A paired *t*-test was applied for within-group comparisons. Log-rank test was applied for comparison between Kaplan–Meier curves. *p* < 0.05 was considered statistically significant.

## 3. Results

### 3.1. Baseline of the Study Population

In total, 187 patients were recruited. The average time of follow-up was 24.57 ± 5.52 months. At the time of inclusion, 59 out of 187 (31.55%) were at CKD stage 1, 61 (32.62%) at CKD2, 46 (24.60%) at CKD 3, 16 (8.56%) at CKD 4, and 5 (2.67%) at CKD 5. For the pathological diagnoses, 69 out of 187 (36.90%) were diagnosed with IgA nephropathy, 35 (18.72%) with membranous nephropathy, 20 (10.70%) with diabetic nephropathy, 16 (8.56%) with a minimal-change disease, 7 (3.74%) with a tubulointerstitial disease, 7 (3.74%) with hypertensive renal disease, 17 (9.09%) with focal segmental glomerulosclerosis, and 29 (15.51%) with other diseases (6 with lupus nephritis, 4 with obesity-associated nephropathy, 2 with thrombotic microangiopathy, 2 with renal amyloidosis, 2 with proliferative glomerulonephritis, 3 with benign renal small-artery sclerosis, 2 with podocytosis, 2 with ANCA-associated vasculitis renal damage, 1 with renal ischemic changes due to severe vascular lesions, 1 with sclerosing nephritis, 1 with hepatitis-B-associated nephritis, 1 with Henoch–Schönlein purpura nephritis, 1 with light-chain proximal tubulopathy, and 1 with IgG4-associated nephropathy). Some of the patients had combined pathological diagnoses, resulting in a sum of pathological diagnoses of over 187.

During the follow-up, three patients died. One of these deaths was due to multi-organ failure (at CKD stage 3 at the time of inclusion), while the causes of the other two were unknown (at CKD stage 2 at the time of inclusion). Six patients initiated renal replacement therapy, of whom two were at CKD stage 5 at the time of inclusion, two were at CKD 4, and two were at CKD 3. Of the remaining patients who did not die or initiate renal replacement therapy, 18 out of 187 (9.63%) had a sustained decrease (measured at least twice, >3 months in between) of eGFR over 25% from baseline, accompanied by a drop in CKD stage. Furthermore, 48 out of 187 (25.67%) had a sustained increase (measured at least twice, >3 months in between) in eGFR over 25% from baseline, which could be considered as a regression of CKD [30]. The total incidence of CKD progression was 25 out of 187 (13.37%). At the time of the last follow-up, 75 out of 187 (40.11%) patients were at CKD stage 1, 55 (29.41%) were at CKD 2, 29 (15.51%) were at CKD 3, 11 (5.88%) were at CKD 4, and 17 (9.09%) were at CKD 5. The incidence of patients’ prognosis over CKD stage 3 or over is 57 out of 187 (30.48%). The baseline characters are in Table 1. The patients’ Scr and eGFR before and after are depicted in Appendix A.

### 3.2. Feature Selections for CKD Prognosis by Lasso Regression

In total, 973 features, including clinical index, pathological changes, ultrasound parameters, and radiomics signature, were entered into the lasso regression analysis. The 26 significant features selected were Scr, eGFR, 24-h urinary protein at baseline, glomerular global sclerosis rate, glomerular focal segmental sclerosis rate, interstitial inflammation, and 20 radiomics signatures. Details of the features and their coefficients are shown in Figure 2.

### 3.3. Cox regression for CKD Prognosis

Univariate and multivariate Cox regression analyses were performed next, using both statistical and clinically significant parameters from the results of the differential comparison (Table 1) and lasso regression (Figure 2). The parameters of sinus wavelet HLH first-order maximum, sinus wavelet HLH first-order range, and medulla wavelet HLH, first-order 10th percentile from ultrasound radiomics, were excluded because their values were all less than 0.0001, and therefore meaningless for clinical use. As shown in Table 2, eGFR, Scr, ACR at baseline, median SWE value of left renal cortex (L_C_median), length of left kidney, and nine radiomics signatures were statistically significant for the secondary outcome. The hypothesis of proportional hazard for the Cox regression is proven in Appendix A.

Continuous features with *p* < 0.1 in the univariate Cox regression analysis were further calculated as cutoff values for the secondary outcome based on Kaplan–Meier method (Appendix A). The cutoff values for eGFR, Scr, ACR, length of left kidney, L_C_median, mean SWE value of left renal sinus (L_S_mean), and the nine radiomics signatures were 51.23 mL/min/1.73 m^2^, 102 μmol/L, 1000 mg/g, 98 mm, 13 kPa, 15.5 kPa, 0.37, 0.16, 9.43, 2.68, 0.08, 0.46, 258.03, 0.25, and 3.66, respectively. They were then divided into a high group or low group and drawn into Kaplan–Meier curves with categorical features of tubular atrophy and artery/arteriole hyalinosis (Figure 3). Except for L_S_mean, all the subgroups of features were shown to be statistically different for the secondary outcome.

To better explain the clinical meaning of the Cox regression model, we built four further Cox regression models using the statistically significant features in a multivariate Cox regression analysis. Model-All used all the features. Model-Clin + Patho used the clinical features of eGFR, Scr, ACR at baseline, pathological features of tubular atrophy, artery/arteriole hyalinosis, and length of left kidney, which are commonly used in clinical practice. Model-Clin + SWE used the clinical features, length of left kidney, and the SWE parameters of L_C_median and L_S_mean. Model-Clin + SWE + Radiomics added the nine radiomics signatures. There is no problem with multicollinearity in our multivariable model according to the multicollinearity diagnosis (single variance inflation factor < 10 and average variance inflation factor < 6 [31,32] in Appendix A. Likelihood chi-square test, C-index, and time-dependent ROC all illustrated that Model-All performed the best, with an average area under curves (AUCs) of time-dependent ROC over 0.9 (Table 3, Appendix A; Figure 4). Model-Clin + SWE + Radiomics improved the prediction ability of Model-Clin + SWE and Model-Clin + Patho.

### 3.4. Nomogram for CKD Prognosis

As shown in Figure 5, a prognostic nomogram was built based on the Cox regression model using all the statistically significant features from multivariate Cox regression analysis. The calibration curve for the 1-year, 2-year, and 2.5-year survival of those with CKD stage over 3 demonstrated the good performance of the prognostic nomogram (Figure 5c–e). The decision curve for the net benefit demonstrated that the nomogram was more reliable at predicting survival after two years (Figure 5f,g).

### 3.5. Predicting Models for CKD Prognosis Using Machine Learning and Deep Learning

As shown in Figure 6, the RSF and Coxboost prediction models performed best in the random-split test dataset (baseline character in Appendix A) among all the machine learning and deep learning models, with C-indices of 0.8095 (0.7938–0.8303), and 0.8139 (0.8037–0.8307). However, compared to the Cox regression model using the same features (Model-All), the machine learning and deep learning models dropped in predictive performance at 30 months.

## 4. Discussion

In this cohort study, we found that L_C_median is an independent risk factor for CKD progression to CKD stage 3 or over by multivariate Cox regression with a hazard ratio of 0.890 (0.796–0.994) (*p* < 0.05, Table 2). This finding might support the early prediction of CKD progression in clinical settings, especially in healthcare centers with the inability to perform kidney biopsies. Patients with a high risk of disease progression according to the nomogram can be treated more aggressively.

Although few studies have reported the predictive value of SWE for the prognosis of CKD in adult native kidneys, our finding still complies with Liu et al.’s finding of higher SWE values for the left and right renal cortex in children’s CKD progression using the same machine [33]. Kennedy et al. also found that the renal cortex stiffness of allograft, reflected by point-shear elastography ultrasound, increased at baseline in those who developed into graft loss later [34]. This finding may be one of the reasons why high intra-subject variability was found by Radulescu et al. among CKD patients [35]. As shown in Appendix A, some patients had worsening eGFR while others at the same CKD stage had stable or recovering eGFR. This was consistent with the insensitivity of eGFR in subclinical injury or initially adaptive repair [3].

Additionally, using SWE, we found that the patients who progressed to or stayed at CKD stage 3–5 did not statistically differ in all SWE parameters from the patients who stayed in or regressed to CKD stages 1–2 at the inclusion time (Table 1). The reasons for this might be confounding factors. Studies have also shown that tissue viscosity might increase shear wave velocity [36,37]. Furthermore, tissue viscosity has been reported as a marker for hepatic necroinflammation [38]. This may also account for the non-linear relationship between SWE value and eGFR or fibrosis [12]. Furthermore, the SWE value in the medulla was more unstable than the value in the cortex because the organized microstructures of the loop of Henle and the vasa recta in the kidney medulla made the shear wave velocity vary according to the direction of measurement [39]. In line with the literature, we found that the SWE (Young’s modulus) value in the cortex is more predictive than that in the medulla or sinus.

In addition, we also found nine radiomics signatures that were independent risk factors for CKD progression (*p* < 0.05). Furthermore, the Cox regression model containing them performed better than the Model-Clin + Patho and Model-Clin + SWE (Figure 4; Table 3, Appendix A). These represent the added value of radiomics in SWE. All of the significant radiomics signatures were from wavelet-transformed images, which offer proven reproducibility [16]. We manually delineated the ROIs for radiomics analysis based on the ROIs of the SWE ultrasound images instead of segmenting the whole renal cortex or sinus, or medulla. Because focal or segmental fibrosis might occur during CKD [40], comparing the features of SWE, the use of radiomics obtained from a very similar part to biopsy is more appropriate. To minimize the effect of the ROI position on the reproducibility of radiomics signatures, we used the median value from four images per patient, similar to the SWE parameters. Machine learning offers advantages in the handling of large-scale and complex clinical data. However, risk bias is receiving attention due to the selection of modeling features and overfitting [15]. Therefore, we built the machine learning models based on the same features as the traditional Cox regression model and minimized overfitting by random-split, 10-fold validation, and Bootstrap. We found that the Cox regression model was no worse than the machine learning models, as was reported previously [41]. The reason for this might be the overfitting of machine learning models in hidden layers or nodes.

The limitations of this study are of its short follow-up time and use of a single center, which may have caused selection bias. External validation and a 5-year or 10-year follow-up will be needed in our feature studies.

## 5. Conclusions

In conclusion, the median SWE value of the left kidney cortex can independently predict a 2.5-year CKD prognosis of CKD stage 3 or over. Radiomics can improve the predictive performance of SWE for CKD progression. Traditional Cox regression modeling is no worse than machine learning.

## Figures and Tables

**Figure 1 diagnostics-12-02678-f001:**
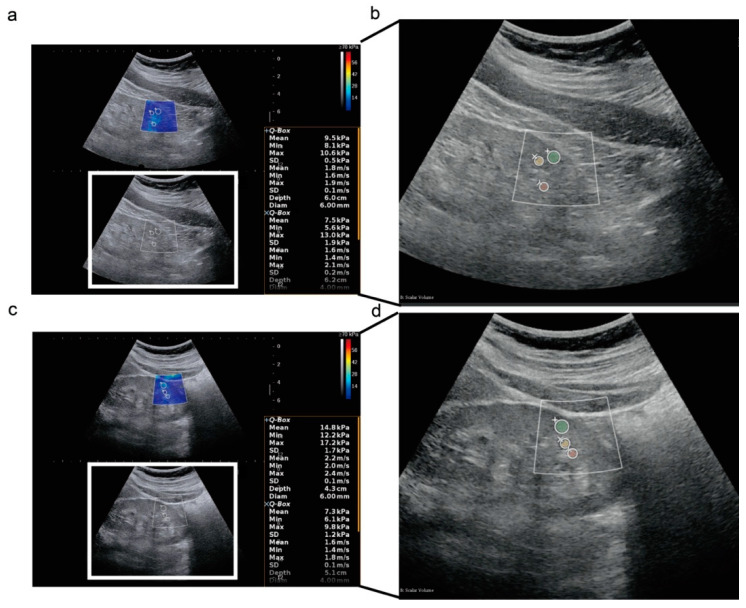
Two-dimensional shear wave elastography ultrasound images in a longitudinal plane. (**a**,**b**) Two-dimensional shear wave elastography ultrasound (2D-SWE) images and radiomics analysis of regions of interest (ROIs) in a 63-year-old female patient at CKD stage 3, who recovered to CKD stage 2 at the last follow-up. The SWE value from the single examination of the left kidney cortex was 9.5 kPa, shown in the right box; (**c**,**d**) 2D-SWE images and radiomics analysis of ROIs in a 28-year-old female patient at CKD stage 3, who progressed to CKD stage 5 at the last follow-up; (+), (Í), (å) represents left kidney cortex, sinus, and medulla, respectively; green, yellow, and red masks in (**b**,**d**) represent the ROI of left kidney cortex, sinus, and medulla, respectively, drawn for radiomics analysis according to the ROIs of 2D-SWE.

**Figure 2 diagnostics-12-02678-f002:**
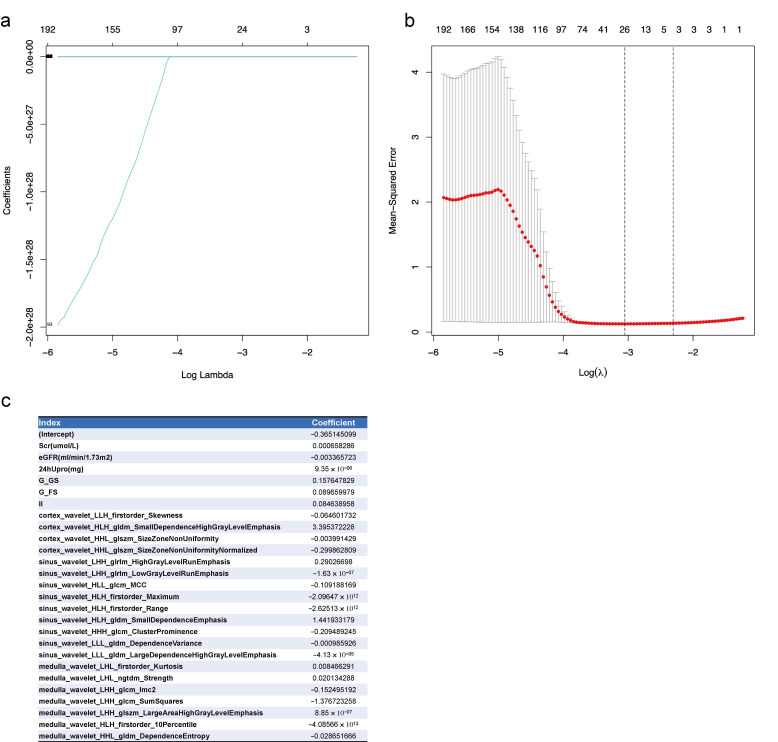
Lasso regression of feature selection from the clinical index, pathological changes, ultrasound parameters, and radiomics signatures for CKD prognosis. (**a**) Lasso coefficients of all features. Log lambda (λ) represents the regularization penalty parameter; (**b**) cross-validation to select the optimal parameter λ. The optimal number of features was 26; the minimum value is represented by red dotted vertical lines. The two dotted gray lines represent the standard deviation. (**c**) The 26 selected features and their coefficients. Scr, serum creatinine at baseline; eGFR, eGFR at baseline; 24hUpro, twenty-four-hour total urinary protein; G_G_Sclerosis, Glomerular_Global Sclerosis; G_FS_Sclerosis, Glomerular_Focal Segmental Sclerosis; II, interstitial inflammation.

**Figure 3 diagnostics-12-02678-f003:**
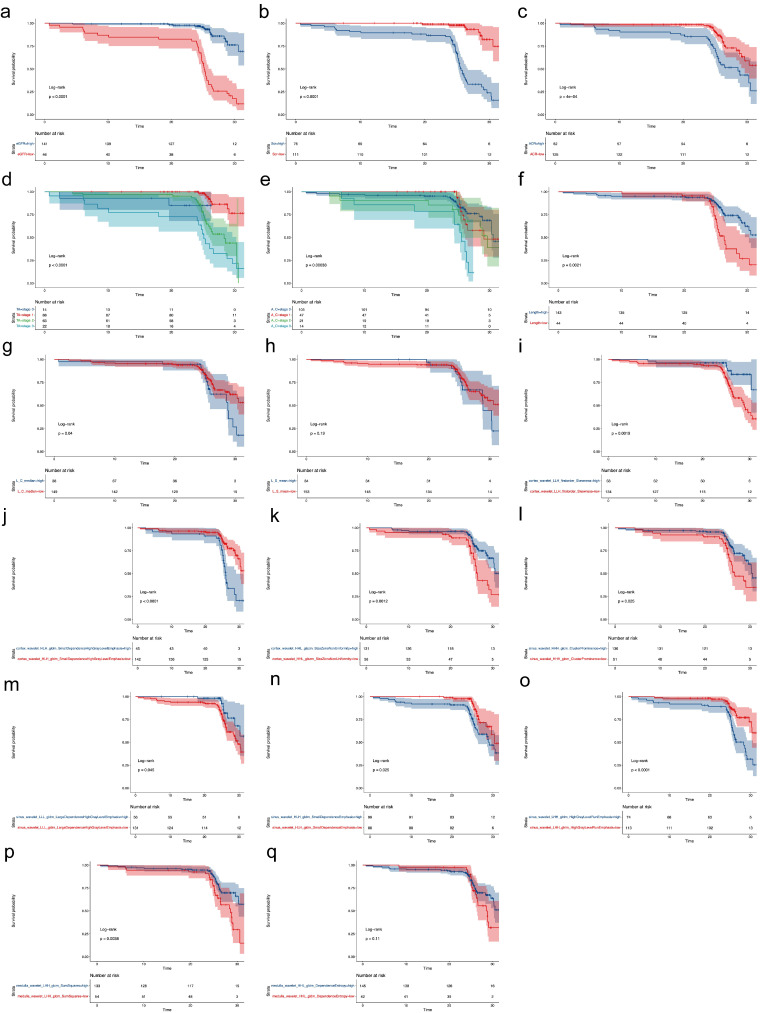
Kaplan–Meier curve for CKD prognosis. The features were divided into a high group or a low group based on their cutoff value. (**a**–**q**), the Kaplan–Meier curve for CKD prognosis by different parameters as shown in the Strata column in each subfigure. The cutoff values for eGFR (**a**), Scr (**b**), ACR (**c**), length of left kidney (**f**), median SWE value of left renal cortex (**g**), mean SWE value of left renal sinus (**h**), cortex wavelet LLH first order Skewness (**i**), cortex wavelet HLH gldm SmallDependenceHighGrayLevelEmphasis (**j**), cortex wavelet HHL glszm SizeZoneNonUniformity (**k**), sinus wavelet LHH glrlm HighGrayLevelRunEmphasis (**l**), sinus wavelet HLH gldm SmallDependenceEmphasis (**m**), sinus wavelet HHH glcm ClusterProminence (**n**), sinus wavelet LLL gldm LargeDependenceHighGrayLevelEmphasis (**o**), medulla wavelet LHH glcm SumSquares (**p**), medulla wavelet HHL gldm DependenceEntropy (**q**) were 51.23 mL/min/1.73 m^2^, 102 μmol/L, 1000 mg/g, 98 mm, 13 kPa, 15.5 kPa, 0.37, 0.16, 9.43, 2.68, 0.08, 0.46, 258.03, 0.25, and 3.66, respectively.

**Figure 4 diagnostics-12-02678-f004:**
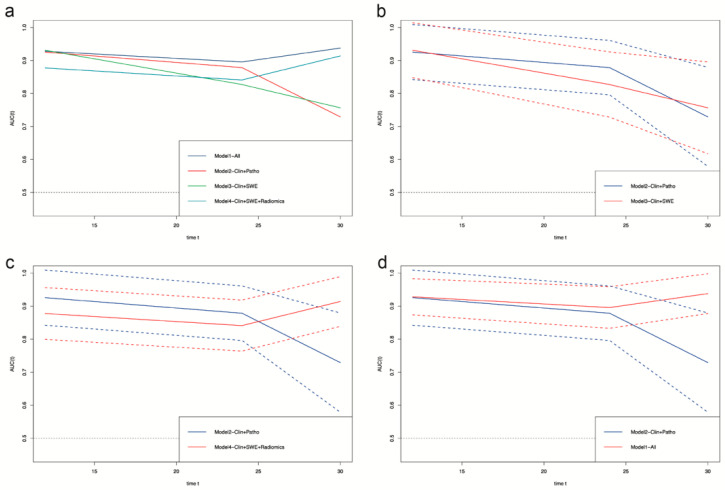
Time-dependent ROC of survival prediction models based on Cox regression. (**a**), the comparison of time-dependent ROC among all four models. (**b**), the comparison of time-dependent ROC between Model2–Clin+Path and Model3–Clin+SWE. (**c**), the comparison of time-dependent ROC between Model2–Clin+Path and Model4–Clin+SWE+Radiomics. (**d**), the comparison of time-dependent ROC between Model2–Clin+Path and Model1–All. Clin, clinical features of eGFR at baseline, Scr at baseline, and ACR at baseline; Patho, pathological features of tubular atrophy and artery/arteriole hyalinosis; SWE, elastography parameters of median SWE value of left renal cortex and mean SWE value of left renal sinus; Radiomics, radiomics signatures of cortex wavelet LLH first order Skewness, cortex wavelet HLH gldm SmallDependenceHighGrayLevelEmphasis, cortex wavelet HHL glszm SizeZoneNonUniformity, sinus wavelet LHH glrlm HighGrayLevelRunEmphasis, sinus wavelet HLH gldm SmallDependenceEmphasis, sinus wavelet HHH glcm ClusterProminence, sinus wavelet LLL gldm LargeDependenceHighGrayLevelEmphasis, medulla wavelet LHH glcm SumSquares, and medulla wavelet HHL gldm DependenceEntropy; Model1-All, Cox regression model of all features; Model2-Clin + Patho, Cox regression model of clinical and pathological features and length of left kidney; Model3-Clin + SWE, Cox regression model of clinical features, length of left kidney, elastography parameters; Model4-Clin + SWE + Radiomics, Cox regression model of clinical features, length of left kidney, elastography parameters, and radiomics signatures. The dashed lines represent a 95% confidence interval.

**Figure 5 diagnostics-12-02678-f005:**
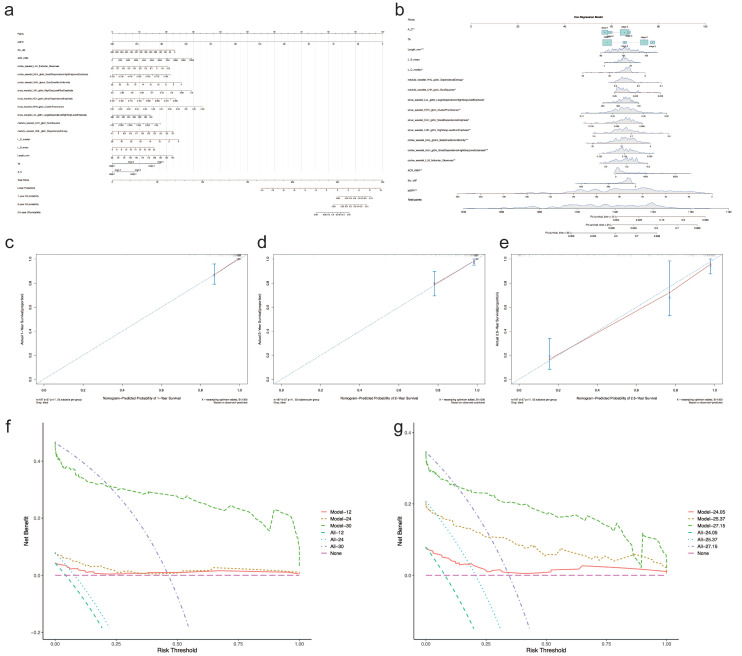
Nomogram for CKD prognosis. (**a**) Prognostic nomogram. (**b**) Dynamic nomogram demonstrating the prediction of overall survival probability by each feature. The risk points (0–100) of each feature can be determined by drawing a vertical line to the line of “Points” on the top from the value of each feature (red dots). A total point was obtained by adding up the risk points of all features. The overall survival probability can be found by drawing a vertical line from the corresponding total point in the line of “Total Points” (the red arrow line). (**c**–**e**) Calibration curve for predicting survival at 1-year, 2-year, and 2.5-year intervals. The red line represents the performance of the nomogram, which is better if it is closer to the diagonal line, where nomogram-predicted probability equals actual survival proportion. B represents times of Bootstrap validation. (**f**,**g**) The decision curve for the nomogram. The pink line annotated as “None” represents the assumption that no patients have progressed to CKD stage 3 or over. The dotted line annotated as “All-xx” represents the assumption that all patients have progressed to CKD stage 3 or over. The further the model line from the “None” or “All” lines, the greater net benefit to the model gets. “Model-1” and “All-12” in (**f**) are at the time of 12 months, “Model-2” and “All-24” are at the time of 24 months, and “Model-3” and “All-30” are at the time of 30 months. The values “24.05”, “25.37”, and “27.15” in (**g**) are the quartiles of total follow-up time. “Model” in (**f**,**g**) all represent the Cox regression model used to build the nomogram.

**Figure 6 diagnostics-12-02678-f006:**
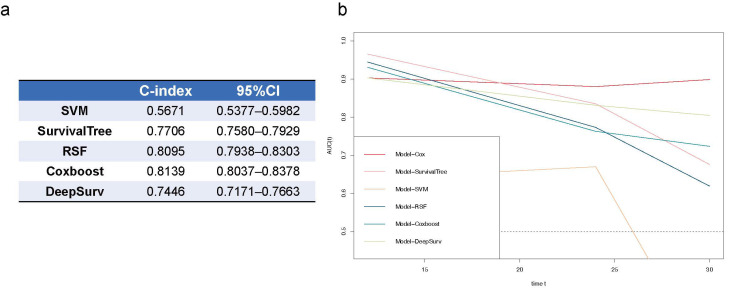
Comparison of survival prediction models for CKD prognosis using Cox regression, machine learning, and deep learning. (**a**) C-index of machine learning models by SVM, SurvivalTree, RSF, Coxboost, and deep learning model of DeepSurv; (**b**) time-dependent ROCs of all prediction models. SVM, support vector machine; RSF, random survival forest; 95%CI, 95% confidence interval.

**Table 1 diagnostics-12-02678-t001:** Baseline characteristics of the study cohort.

	**Total (*n* = 187)**	**CKD fo_1~2 (*n* = 130)**	**CKD fo_3~5 (*n* = 57)**	***p* Value**
Age(year)	45.00 (32.00–59.00)	40.00 (30.00–40.00)	52.00 (38.50–52.00)	<0.001
Sex (male%)	105 (56.1%)	67 (63.80%)	38 (36.20%)	0.055
BMI (kg/m^2^)	24.30 (21.96–27.16)	24.19 (21.98–24.19)	24.80 (21.80–24.80)	0.426
SBP (mmHg)	141.00 (121.00–165.00)	135.50 (119.00–135.50)	157.00 (128.00–157.00)	0.003
DBP (mmHg)	77.50 (70.00–85.75)	77.50 (67.75–77.50)	77.00 (72.00–77.00)	0.362
eGFR (MDRD) (mL/min/1.73 m^2^)	73.35 (51.96–101.88)	86.55 (69.04–86.55)	39.01 (24.94–39.01)	<0.001
Scr (μmol/L)	92.00 (68.00–131.00)	81.00 (62.00–81.00)	165.00 (117.50–165.00)	<0.001
BUN (mmol/L)	5.50 (4.23–7.38)	4.70 (3.90–4.70)	7.90 (5.80–7.90)	<0.001
UA (μmol/L)	363.50 (299.50–412.50)	351.50 (281.25–351.50)	389.00 (343.00–389.00)	0.001
Alb (g/L)	35.60 (29.95–41.05)	36.45 (29.11–36.45)	35.00 (30.25–35.00)	0.612
24hUpro (mg)	1454.40 (624.60–3397.15)	1119.15 (583.40–1119.15)	2493.00 (1186.80–2493.00)	<0.001
ACR (mg/g)	565.95 (253.03–1788.85)	421.85 (193.65–421.85)	1084.40 (441.00–1084.40)	<0.001
**Pathological Changes**	**Total (*n* = 187)**	**CKD fo_1~2 (*n* = 130)**	**CKD fo_3~5 (*n* = 57)**	***p* Value**
G_G_Sclerosis (%)	20.00% (5.88–43.75%)	12.77% (0.00–28.57%)	46.67% (25.83–62.02%)	<0.001
G_FS_Sclerosis (%)	0.00% (0.00–7.69%)	0.00% (0.00–6.47%)	3.23% (0.00–11.81%)	0.042
G_Crescents (%)	0.00% (0.00–3.33%)	0.00% (0.0–3.54%)	0.00% (0.00–4.41%)	0.959
G_Fibrinoid necrosis (%)	0.00% (0.00–0.00%)	0.00% (0.00%–0.00%)	0.00% (0.00–0.00%)	0.290
Mesengial matrix hyperplasia				<0.001
0	17 (9.10%)	10 (58.80%)	7 (41.20%)	
1	140 (74.90%)	105 (75.00%)	35 (25.00%)	
2	20 (10.70%)	14 (70.00%)	6 (30.00%)	
3	10 (5.30%)	1 (10.00%)	9 (90.00%)	
Mesangial hypercellularity				0.096
0	30 (16.00%)	19 (63.30%)	11 (36.70%)	
1	134 (71.70%)	99 (73.90%)	35 (26.10%)	
2	22 (11.80%)	12 (54.50%)	10 (45.50%)	
3	1 (0.50%)	0 (0.00%)	1 (100.00%)	
Intra–capillary proliferation				0.069
0	144 (77.00%)	102 (70.80%)	42 (29.20%)	
1	2 (1.10%)	1 (50.00%)	1 (50.00%)	
2	41 (21.90%)	27 (65.90%)	14 (34.10%)	
3	0 (0.00%)	0 (0.00%)	0 (0.00%)	
Capillary wall hyalinosis				0.738
0	114 (61.00%)	82 (71.90%)	32 (28.10%)	
1	49 (26.20%)	31 (63.30%)	18 (36.70%)	
2	21 (11.20%)	15 (71.40%)	6 (28.60%)	
3	3 (1.60%)	2 (66.70%)	1 (33.30%)	
Tubular atrophy				<0.001
0	14 (7.50%)	12 (85.70%)	2 (14.30%)	
1	88 (47.10%)	78 (88.60%)	10 (11.40%)	
2	63 (33.70%)	37 (58.70%)	26 (41.30%)	
3	22 (11.80%)	3 (13.60%)	19 (86.40%)	
Interstitial inflammation				<0.001
0	14 (7.50%)	13 (92.90%)	1 (7.10%)	
1	91 (48.70%)	82 (90.10%)	9 (9.90%)	
2	60 (32.10%)	33 (55.00%)	27 (45.00%)	
3	22 (11.80%)	2 (9.10%)	20 (90.90%)	
Interstitial fibrosis				<0.001
0	13 (7.00%)	12 (92.30%)	1 (7.70%)	
1	92 (49.20%)	81 (88.00%)	11 (12.00%)	
2	61 (32.60%)	34 (55.70%)	27 (44.30%)	
3	21 (11.20%)	3 (14.30%)	18 (85.70%)	
Artery/arteriole intima thickening				0.114
0	79 (42.20%)	59 (74.70%)	20 (25.30%)	
1	39 (20.90%)	30 (76.90%)	9 (23.10%)	
2	59 (31.60%)	34 (57.60%)	25 (42.40%)	
3	10 (5.30%)	7 (70.00%)	3 (30.00%)	
Artery/arteriole hyalinosis				<0.001
0	105 (56.10%)	81 (77.10%)	24 (22.90%)	
1	47 (25.10%)	35 (74.50%)	12 (25.50%)	
2	21 (11.20%)	10 (47.60%)	11 (52.40%)	
3	14 (7.50%)	4 (28.60%)	10 (71.40%)	
Grade of chronic changes				<0.001
Minimal (0–1)	10 (5.30%)	10 (100.00%)	0 (0.00%)	
Mild (2–4)	66 (35.30%)	59 (89.40%)	7 (10.60%)	
Moderate (5–7)	61 (32.60%)	47 (77.00%)	14 (23.00%)	
Severe (≥8)	50 (26.70%)	14 (28.00%)	36 (72.00%)	
**Ultrasound**	**Total (*n* = 187)**	**CKD fo_1~2 (*n* = 130)**	**CKD fo_3~5 (*n* = 57)**	***p* Value**
L_C_mean (kPa)	10.60 (9.50–12.50)	10.35 (9.30–11.95)	10.80 (9.65–13.30)	0.126
L_C_median (kPa)	10.60 (9.20–12.60)	10.50 (9.18–12.10)	10.80 (9.30–13.40)	0.174
L_M_mean (kPa)	6.50 (5.50–8.20)	6.45 (5.28–8.13)	6.80 (5.70–8.70)	0.128
L_M_median (kPa)	6.50 (5.30–8.40)	6.35 (5.20–8.40)	6.90 (5.80–8.75)	0.124
L_S_mean (kPa)	13.40 (12.10–14.80)	13.10 (11.90–14.58)	13.80 (11.85–15.25)	0.305
L_S_median (kPa)	13.40 (11.90–14.90)	13.10 (11.70–14.60)	13.90 (11.80–15.35)	0.235
Length (mm)	106.00 (99.00–112.00)	106.00 (101.00–112.25)	103.00 (95.50–110.00)	0.037
Width (mm)	45.00 (42.00–49.00)	45.00 (42.00–48.00)	45.00 (40.50–49.00)	0.904
Thickness (mm)	43.40 (39.00–46.50)	44.00 (40.30–47.00)	42.00 (38.50–45.50)	0.093
Kidney volume (cm^3^)	201.35 (173.04–238.66)	205.95 (179.58–239.10)	188.93 (158.01–237.50)	0.159

CKD fo_1~2, patients at CKD stage 1~2 at the last follow-up time. CKD fo_3~5, patients at CKD stage 3~5 at the last follow-up time; G_G_Sclerosis, Glomerular_Global Sclerosis; G_FS_Sclerosis, Glomerular_Focal Segmental Sclerosis; G_Crescents, Glomerular_Crescents; G_Fibrinoid necrosis, Glomerular_Fibrinoid necrosis; L_C_mean, mean SWE value of left renal cortex; L_C_median, median SWE value of left renal cortex; L_M_mean, mean SWE value of left renal medulla; L_M_median, median SWE value of left renal medulla; L_S_mean, mean SWE value of left renal sinus; L_S_median, median SWE value of left renal sinus; Length, Width, Thickness, and Kidney volume, are length, width, thickness, and product of length and width and thickness of left kidney, respectively. All parameters were collected at the time of biopsy.

**Table 2 diagnostics-12-02678-t002:** Univariate and multivariate Cox regression for CKD prognosis.

Parameters		Univariate		Multivariate	
		HR (95%CI)	*p* Value	HR (95%CI)	*p* Value
**Clinical Index**	Age (year)	0.953 (0.904–1.003)	0.066		
	SBP (mmHg)	1.002 (0.995–1.009)	0.613		
	eGFR (mL/min/1.73 m^2^)	0.865 (0.806–0.93)	<0.001	0.927 (0.9–0.955)	<0.001
	Scr (μmol/L)	0.978 (0.965–0.992)	0.002	0.994 (0.989–0.999)	0.013
	BUN (mmol/L)	1.195 (0.957–1.492)	0.116		
	UA (μmol/L)	0.997 (0.987–1.006)	0.504		
	24hUpro (mg)	0.999998 (0.999767–1.000229)	0.985		
	ACR (mg/g)	1.001 (1.001–1.002)	<0.001	1.000489 (1.000247–1.001)	<0.001
**Pathological** **Changes**	G_GS (%)	7.53 (0.311–182.252)	0.214		
	G_FS (%)	5.63 (0.152–209.022)	0.349		
	M_M (1)	39.261 (1.175–1311.474)	0.040		
	M_M (2)	0.669 (0.009–52.184)	0.857		
	M_M (3)	6.067 (0.268–137.282)	0.257		
	TA (1)	0.002 (0.000009–0.649)	0.035	0.316 (0.038–2.602)	0.284
	TA (2)	6.465 (0.018–2293.191)	0.533	3.584 (0.435–29.553)	0.236
	TA (3)	1.524 (0.000111–20913.596)	0.931	6.173 (0.553–68.902)	0.139
	II (1)	0.348 (2.614 × 10^–27^–4.638 × 10^25^)	0.973		
	II (2)	15,636.424 (1.029 × 10^–22^–2.374 × 10^30^)	0.754		
	II (3)	5.148 (2.995 × 10^–26^–8.847 × 10^26^)	0.958		
	IF (1)	5677.54 (3.1539 × 10^–23^–1.022 × 10^30^)	0.779		
	IF (2)	0.29 (1.641 × 10^–27^–5.126 × 10^25^)	0.968		
	IF (3)	55,447.23 (2.382 × 10^–22^–1.290 × 10^31^)	0.724		
	A_C (1)	0.092 (0.010–0.806)	0.031	0.273 (0.109–0.682)	0.005
	A_C (2)	0.122 (0.016–0.922)	0.041	0.385 (0.122–1.217)	0.104
	A_C (3)	7.482 (0.883–63.383)	0.065	1.317 (0.464–3.741)	0.605
	Grade of chronic changes (1)	17,027.838 (9.106 × 10^–35^–3.184 × 10^42^)	0.828		
	Grade of chronic changes (2)	152.155 (8.861 × 10^–37^–2.613 × 10^40^)	0.911		
	Grade of chronic changes (3)	6.777 (3.582 × 10^–38^–1.282 × 10^39^)	0.966		
**Ultrasound** **Parameters**	L_C_mean (kPa)	0.281 (0.073–1.08)	0.065		
	L_C_median (kPa)	3.236 (0.945–11.076)	0.061	0.890 (0.796–0.994)	0.039
	L_M_mean (kPa)	1.457 (0.329–6.452)	0.620		
	L_M_median (kPa)	0.569 (0.143–2.263)	0.423		
	L_S_mean (kPa)	3.024 (0.875–10.449)	0.080	1.154 (0.971–1.371)	0.104
	L_S_median (kPa)	0.444 (0.159–1.243)	0.122		
	Length (mm)	1.267 (1.117–1.437)	<0.001	1.077 (1.034–1.122)	<0.001
	Kidney_volume (cm^2^)	0.995 (0.977–1.014)	0.625		
**Ultrasound** **Radiomics**	cortex_wavelet_LLH_firstorder_Skewness	0.000363 (0.000002–0.056)	0.002	0.032 (0.003–0.311)	0.003
	cortex_wavelet_HLH_gldm_ SmallDependenceHighGrayLevelEmphasis	2.005 × 10^66^ (1.482 × 10^29^–2.714 × 10^103^)	<0.001	7.876 × 10^23^ (1.672 × 10^11^–3.709 × 10^36^)	<0.001
	cortex_wavelet_HHL_glszm_ SizeZoneNonUniformity	0.704 (0.536–0.925)	0.012	0.789 (0.691–0.902)	0.001
	cortex_wavelet_HHL_glszm_ SizeZoneNonUniformityNormalized	1811.003 (0.000122–2.698 × 10^10^)	0.373		
	sinus_wavelet_LHH_glrlm_ HighGrayLevelRunEmphasis	916,226,561.138 (300.893–2.790 × 10^15^)	0.007	206,763.534 (38.345–1.115 × 10^10^)	0.005
	sinus_wavelet_LHH_glrlm_ LowGrayLevelRunEmphasis	a			
	sinus_wavelet_HLL_glcm_MCC	129.179 (0.002–7,675,218.559)	0.386		
	sinus_wavelet_HLH_gldm_ SmallDependenceEmphasis	5.848 × 10^–45^ (7.627 × 10^–77^–4.483 × 10^–13^)	0.007	7.163 × 10^–25^ (4.143 × 10^–44^–0.000012)	0.014
	sinus_wavelet_HHH_glcm_ClusterProminence	1.774 × 10^46^ (9.389 × 10^11^–3.350 × 10^80^)	0.008	9.179 × 10^21^ (55,633,770.994–1.514 × 10^36^)	0.002
	sinus_wavelet_LLL_gldm_ DependenceVariance	0.773 (0.184–3.243)	0.725		
	sinus_wavelet_LLL_gldm_ LargeDependenceHighGrayLevelEmphasis	0.985 (0.972–0.998)	0.023	0.993 (0.987–0.999)	0.022
	medulla_wavelet_LHL_firstorder_Kurtosis	1.800 (0.793–4.086)	0.160		
	medulla_wavelet_LHL_ngtdm_Strength	1.685 (0.491–5.789)	0.407		
	medulla_wavelet_LHH_glcm_Imc2	5580.503 (0–3.695 × 10^11^)	0.348		
	medulla_wavelet_LHH_glcm_SumSquares	0 (0–5.364 × 10^–220^)	<0.001	5.146 × 10^–119^ (2.670 × 10^–202^–9.917 × 10^–36^)	0.005
	medulla_wavelet_LHH_glszm_ LargeAreaHighGrayLevelEmphasis	1.002 (0.999769–1.004)	0.082		
	medulla_wavelet_HHL_gldm_ DependenceEntropy	0.001 (0.000003–0.615)	0.034	0.024 (0.001–0.691)	0.030

Multivariate Cox regression analysis by the method of stepwise-backward used all parameters with likelihood-ratio test *p* < 0.1 in univariate analysis. a, The degree of freedom is reduced because of constant or linearly dependent covariates. Pathological changes were categorical covariates; the first category was set for reference. G_GS, Glomerular_Global Sclerosis. G_FS, Glomerular_Focal Segmental Sclerosis. Mesangial matrix hyperplasia (1), Mesangial matrix hyperplasia grade 1 vs. 0. Mesangial matrix hyperplasia (2), Mesangial matrix hyperplasia grade 2 vs. 0. Mesangial matrix hyperplasia (3), Mesangial matrix hyperplasia grade 3 vs. 0. Other pathological changes were similar: TA, tubular atrophy; II, interstitial inflammation; IF, interstitial fibrosis; A_C, artery/arteriole hyalinosis; L_C_mean, mean SWE value of left renal cortex; L_C_median, median SWE value of left renal cortex; L_M_mean, mean SWE value of left renal medulla; L_M_median, median SWE value of left renal medulla; L_S_mean, mean SWE value of left renal sinus; L_S_median, median SWE value of left renal sinus; Length, length of the left kidney; kidney volume, the product of length, width, and thickness of left kidney. HR, hazard ratio; CI, confidence interval.

**Table 3 diagnostics-12-02678-t003:** Comparison of Cox regression models.

Cox Regression Model	L.R.Chisq	*p*-Value
Model-Clin + Patho vs. Model-Clin + SWE	7.2347	0.1240
Model-Clin + Patho vs. Model-Clin + SWE + Radiomics	27.0800	0.0001
Model-Clin + Patho vs. Model-All	63.9548	<0.0001
Model-Clin + SWE + Radiomics vs. Model-All	36.8748	<0.0001

L.R.Chisq, likelihood chi-square test; Clin, clinical features of eGFR at baseline, Scr at baseline, ACR at baseline; Patho, pathological features of tubular atrophy, artery/arteriole hyalinosis; SWE, elastography parameters of median SWE value of left renal cortex, mean SWE value of left renal sinus; Radiomics, radiomics signatures of cortex wavelet LLH first order Skewness, cortex wavelet HLH gldm SmallDependenceHighGrayLevelEmphasis, cortex wavelet HHL glszm SizeZoneNonUniformity, sinus wavelet LHH glrlm HighGrayLevelRunEmphasis, sinus wavelet HLH gldm SmallDependenceEmphasis, sinus wavelet HHH glcm ClusterProminence, sinus wavelet LLL gldm LargeDependenceHighGrayLevelEmphasis, medulla wavelet LHH glcm SumSquares, and medulla wavelet HHL gldm DependenceEntropy; Model1-All, Cox regression model of all features; Model2-Clin + Patho, Cox regression model of clinical and pathological features and length of left kidney; Model3-Clin + SWE, Cox regression model of clinical features, length of left kidney, elastography parameters; Model4-Clin + SWE + Radiomics, Cox regression model of clinical features, length of left kidney, elastography parameters, and radiomics signatures.

## Data Availability

All data mentioned in this manuscript is available with publication upon reasonable request through emails to the correspondence author Shan Mou, shan_mou@126.com.

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
