# Peer review of "Predicting Progression of Kidney Injury Based on Elastography Ultrasound and Radiomics Signatures"

_diagnostics, 2022, doi:10.3390/diagnostics12112678_

Round 1
Reviewer 1 Report
Zhu M et al. examined the prognostic ability of shear wave elastography (SWE) on kidney outcomes in a prospective study. The study included a total of 187 patients with a mean observation time of 24.5 months. The results showed the L_C_median was associated with CKD progression to eGFR<60 mL/min/1.73m2 independent of kidney function and pathological variables. The advantages of the current study are prospective design and non-invasive and quantitative methods by SWE. However, there are some concerns in the manuscript.
1. Please provide the incidence of CKD progression during the follow-up term.
2. Were the patients with rapidly progressive glomerulonephritis (RPGN) excluded?
3. In the legend for Table1, “at the last follow-up time” should be “at the time of kidney biopsy”
4. In Table 2, there may be a concern for multicollinearity since both Scr and eGFR were included in the multivariable model.
5. Please discuss what is observed by SWE.
Author Response
Dear Reviewer,
Thank you very much for your comments. Those comments are all valuable and very helpful for revising and improving our paper, as well as the important guiding significance to our researches. We have studied comments carefully and have made corrections which we hope to make our paper more acceptable. For the English language, we have carefully checked and revised it. We have also used the language editing service based on our revision. Revised portions are marked in red on the paper. Responses to the comments were addressed point by point as below:
Point 1: Please provide the incidence of CKD progression during the follow-up term.
Response 1: Thank you very much for your comments, we have added the incidence of CKD progression, defined according to the 2012 KDIGO Guideline, in Lines 176-186 as “During the follow-up, three patients died; one of these deaths was due to multi-organ failure (at CKD stage 3 at the time of inclusion), while the causes of the other two were unknown (at CKD stage 2 at the time of inclusion). Six patients initiated renal replacement therapy, of whom two were at CKD stage 5 at the time of inclusion, two were at CKD 4, and two were at CKD 3. Of the remaining patients who did not die or initiate renal replacement therapy, 18 out of 187 (9.63%) had a sustained decrease (measured at least twice, >3 months in between) of eGFR over 25% from baseline, accompanied by a drop in CKD stage. Furthermore, 48 out of 187 (25.67%) had a sustained increase (measured at least twice, >3 months in between) in eGFR over 25% from baseline, which could be considered as a regression of CKD [30].The total incidence of CKD progression was 25 out of 187 (13.37%).” We have also added the definition of CKD progression in Line 93-96 in the Materials and Methods part as “CKD progression was defined, according to the 2012 KDIGO Guideline [25], as a sustained decrease (measured at least twice, with >3 months in between) of eGFR over 25% from baseline accompanied by a drop in CKD stage.”. The revisions were marked in red using the “Track Changes” function in MS word for a clearer view.
Point 2: Were the patients with rapidly progressive glomerulonephritis (RPGN) excluded?
Response 2: Thank you very much for your comments, one of the exclusion criteria of our study is rapidly progressive glomerulonephritis (RPGN). We excluded them because of its especially poor prognosis with a relatively very low prevalence (around 2% of the population) which might be different from other types of CKD in etiology, mechanism, kidney composition, and subsequent SWE value. Moreover, in the included 228 biopsy patients who could cooperate with SWE examination, there was no pathological diagnosis of rapidly progressive glomerulonephritis (RPGN) (extensive crescent formation, usually over 50%) according to the definition [PMID:3287904].
For better understanding of our cohort, we have added the detailed pathological diagnosis of patients in Line 163-175 in the text as “For the pathological diagnoses, 69 out of 187 (36.90%) were diagnosed with IgA nephropathy, 35 (18.72%) with membranous nephropathy, 20 (10.70%) with diabetic nephropathy, 16 (8.56%) with minimal-change disease, 7 (3.74%) with tubulointerstitial disease, 7 (3.74%) with hypertensive renal disease, 17 (9.09%) with focal segmental glomerulosclerosis, and 29 (15.51%) with other diseases (6 with lupus nephritis, 4 with obesity-associated nephropathy, 2 with thrombotic microangiopathy, 2 with renal amyloidosis, 2 with proliferative glomerulonephritis, 3 with benign renal small-artery sclerosis, 2 with podocytosis, 2 with ANCA-associated vasculitis renal damage, 1 with renal ischemic changes due to severe vascular lesions, 1 with sclerosing nephritis, 1 with hepatitis-B-associated nephritis, 1 with Henoch–Schönlein purpura nephritis, 1 with light-chain proximal tubulopathy, and 1 with IgG4-associated nephropathy). Some of the patients had combined pathological diagnoses, resulting in a sum of pathological diagnoses over 187”. The revisions were marked in red using the “Track Changes” function in MS word for a clearer view.
Point 3: In the legend for Table1, “at the last follow-up time” should be “at the time of kidney biopsy”
Response 3: Thank you very much for your comments, the groups CKD fo_1-2 and CKD fo_3-5 in Table 1 represent corresponding CKD stages at the last follow-up time. The characteristics parameters in Table 1 were collected at the time of kidney biopsy. Because the secondary outcome of the study is CKD stage over 3 (eGFR <60ml/1.73m2). We did the comparison of baseline characters (parameters) between patients with CKD 1-2 and patients with CKD 3-5 at the last follow-up time in order to select potential risk factors/confounding factors for the outcome for further regression analysis. We have added the further explanation in the legend in Line 263 as “All parameters were collected at the time of biopsy” using the “Track Changes” function in MS word.
Point 4: In Table 2, there may be a concern for multicollinearity since both Scr and eGFR were included in the multivariable model.
Response 4: Thank you very much for your comments, we have added multicollinearity diagnosis for the parameters in the multivariate cox regression (Table S2). When the variance inflation factor (VIF) of one variable from multicollinearity diagnosis is over 10, or the average VIF of all variables is over 6, the variables are in a high probability of multicollinearity [1,2].
[Reference:
1.Vatcheva KP, Lee M, McCormick JB, Rahbar MH. Multicollinearity in Regression Analyses Conducted in Epidemiologic Studies. Epidemiology (Sunnyvale). 2016 April;6(2).
- Hocking RR.; Methods and Applications of Linear Models, 3rd ed.; Wiley: New York, USA, 2013; pp. 142-178]
As shown in Table S1, the VIF of Scr or eGFR is not over 6. Nor are the other parameters. Therefore, there is no problem with multicollinearity in our multivariable model.
We have also added the rationale in Line 225-227 Results part as “There is no problem with multicollinearity in our multivariable model according to the multicollinearity diagnosis (single variance inflation factor <10 and average variance inflation factor <6 [31-32] in Table S2”. The revisions were marked in red using the “Track Changes” function in MS word for a clearer view.
Point 5: Please discuss what is observed by SWE.
Response 5: Thank you very much for your comments, we have added discussion about what had been observed by SWE in Lines 360-371 in the Discussion part, as “Additionally, using SWE, we found that the patients who progressed to or stayed at CKD stage 3–5 did not statistically differ in all SWE parameters from the patients who stayed in or regressed to CKD stages 1–2 at the inclusion time (Table 1). The reasons for this might be the confounding factors. Studies have also shown that tissue viscosity might increase shear wave velocity [36,37]. Furthermore, tissue viscosity has been reported as a marker for hepatic necroinflammation [38]. This may also account for the non-linear relationship between SWE value and eGFR or fibrosis [12]. Furthermore, the SWE value in the medulla was more unstable than the value in the cortex because the organized microstructures of the loop of Henle and the vasa recta in the kidney medulla made the shear wave velocity vary according to the direction of measurement [39]. In line with the literature, we found that the SWE (Young's modulus) value in the cortex is more predictive than that in the medulla or sinus.” We have also marked the changes in red using the “Track Changes” function in the text for a clearer view.
Once again, thank you very much for your valuable comments and suggestions.

Reviewer 2 Report
The manuscript entitled “Predicting progression of kidney injury based on elastography ultrasound and radiomics signatures” submitted by Zhu et al. evaluated the elastography ultrasound and radiomics signatures based signatures for predicting the progression of kidney injury. The authors recruited 130 CKD patients; 130 were from CKD grades 1-2, whereas 57 suffered from CKD grades 2-5. The findings are significant. However, some modifications should be required.
1. Abstract
- The full forms of some abbreviations like SVM, RSF should be provided at their first appearance in the abstract.
2. Introduction
- should be elaborated by discussing the different parameters and markers of CKD, elastography ultrasound and radiomics signatures with relevant citations.
3. Materials and Methods
- The 3. According to the citation (Stevens et al.), CKD stages were not provided. eGFR values should be
CKD1≥90, CDK2= 60-89, CKD3=30-59, CKD4=15-29, CKD5<15.
4. Results
- According to the text (lines 139-142), CKD1+CKD2= 120 patients, CKD3-5= 57, But according to table 1 and lines 148-149, CKD1+CKD2 should be 130.
- For pathological diagnosis, 69 (36.90%) were diagnosed of IgA nephropathy……..(lines 140-144), here, the % values looked not correct. IgA nephropathy, 69 out of 130, so the % should be 63.08. Similar changing required in other values in the mentioned lines.
- the level of significant values mentioned in table 1-3, should not be zero; it may be <0.01, <0.001 and <0.0001 etc.
- Absent of supplementary figures that are required for evaluation of the manuscript.
Author Response
Dear Reviewer,
Thank you very much for your comments. Those comments are all valuable and very helpful for revising and improving our paper, as well as the important guiding significance to our researches. We have studied the comments carefully and have made corrections which we hope to make our paper more acceptable. For the English language, we have carefully checked and revised it. We have also used the language editing service based on our revision. All revised portions are marked in red in the paper using the “Track Changes” function in MS word. Responses to the comments were addressed point by point as below:
Point 1: Abstract - The full forms of some abbreviations like SVM, RSF should be provided at their first appearance in the abstract.
Response 1: Thank you very much for your comments, we have added the full forms of abbreviations at their first appearance in the abstract. The revision is in Lines 24-26 as “……no worse than machine learning models of Support Vector Machine (SVM), SurvivalTree, Random survival forest (RSF), Coxboost, and Deepsurv”. We have also marked the changes in red using the “Track Changes” function in the text for a clearer view.
Point 2: Introduction - should be elaborated by discussing the different parameters and markers of CKD, elastography ultrasound and radiomics signatures with relevant citations.
Response 2: Thank you very much for your comments. For different parameters and markers of CKD,
we have added discussions in Lines 38-40 in the Introduction part, as “However, current monitoring methods for the progression of kidney disease are not ideal. These include biopsy (which is too invasive to repeat), eGFR (which is elevated only after most kidney cells lose regenerative capacity and is insensitive to CKD progression), and proteinuria (which is largely affected by etiology, and insensitive and non-specific to CKD progression) [3-5]. Studies on imaging techniques and urinary biomarkers are emerging as part of the search for promising non-invasive monitoring methods”.
For parameters of elastography ultrasound, we have added discussions in Lines 47-51 in the Introduction part, as “Based on the physical theory that shear wave propagation velocity is higher in stiffer tissues, the stiffness of kidneys can be estimated by the linear formula of Young's modulus using shear wave velocity, obtained through SWE [9,10]. Due to the anisotropy of the kidneys, the SWE parameters usually include the Young's modulus value in the cortex and the Young's modulus in the medulla [11].”
For parameters of radiomics signatures, we have added discussions in Lines 60-68 in the Introduction part, as “PyRadiomics can extract high-throughput quantitative features from the region of interest (ROI) in medical images, including ultrasound images [16-18]. The extracted signatures by PyRadiomics include classes of first-order statistics (19 features), shape descriptors (including 2D and 3D, not often used in ultrasound), and texture classes of gray level cooccurrence matrix (glcm, 24 features), gray level run length matrix (glrlm, 16 features), gray level size zone matrix (glszm,16features), gray level dependence Matrix(Gldm, 14 features), and neighboring gray tone difference Matrix(Ngtdm, 5 features), based on original images or preprocessed images using built-in filters”. We have also marked the changes in red using the “Track Changes” function in the text for a clearer view.
Point 3: Materials and Methods - The 3. According to the citation (Stevens et al.), CKD stages were not provided. eGFR values should be CKD1≥90, CDK2= 60-89, CKD3=30-59, CKD4=15-29, CKD5<15.
Response 3: Thank you very much for your comments. The 5-stage classification of CKD originates from an earlier version of the National Kidney Foundation Practice Guidelines (2003 version, Levey et al. PMID 12859163). The current citation of Stevens et al. (the 2012 version of National Kidney Foundation Practice Guidelines) further classified the stages into 6 by dividing CKD3 into G3a (eGFR=45–59) and G3b (eGFR=30–44). And we agree that eGFR values should be CKD1≥90, CDK2= 60-89, CKD3=30-59, CKD4=15-29, CKD5<15 according to the National Kidney Foundation Practice Guidelines (2003 version).
We have revised this in Line 91-93 in the Material and Methods part as “CKD stage was evaluated based on eGFR (MDRD) as CKD 1, ≥90; CKD 2, 60–89; CKD 3, 30–59; CKD 4, 15–29; and CKD 5, <15 at the time of inclusion and follow-up [24].” and Line 475-476 as “16. Levey AS, Coresh J, Balk E, Kausz AT, Levin A, Steffes MW, et al. National Kidney Foundation practice guidelines for chronic kidney disease: evaluation, classification, and stratification. Ann Intern Med. 2003;139(2):137-47” using the “Track Changes” function for a clearer view.
We have also checked the classification of patients at inclusion time and the last follow-up time. No patients’ classification was affected due to the update of definition. All other references in the manuscript have also been checked.
Point 4: Results - According to the text (lines 139-142), CKD1+CKD2= 120 patients, CKD3-5= 57, But according to table 1 and lines 148-149, CKD1+CKD2 should be 130.
Response 4: Thank you very much for your comments, the CKD stages in the previous Line 139-142 represent patients’ CKD stages at the time of inclusion. The CKD stages in table 1 and previous line 148-149 represents patients’ CKD stages at the time of the last follow-up. As shown in Figure S3, there are patients who had CKD progression, and patients who had CKD regression (a sustained increase (measured at least twice, >3 months in between) of eGFR over 25% from baseline defined according to literature PMID 34100938).
Because the secondary outcome of the study is CKD stage over 3 (eGFR <60ml/1.73m2). We did the comparison of baseline characters/parameters (collected at the time of inclusion ) between patients with CKD 1-2 and patients with CKD 3-5 at the last follow-up time in order to select potential risk factors/confounding factors for the outcome for further regression analysis.
To avoid ambiguity, we have added further explanation in Lines 176-189 in the Results part, as “During the follow-up, three patients died; one of these deaths was due to multi-organ failure (at CKD stage 3 at the time of inclusion), while the causes of the other two were unknown (at CKD stage 2 at the time of inclusion). Six patients initiated renal replacement therapy, of whom two were at CKD stage 5 at the time of inclusion, two were at CKD 4, and two were at CKD 3. Of the remaining patients who did not die or initiate renal replacement therapy, 18 out of 187 (9.63%) had a sustained decrease (measured at least twice, >3 months in between) of eGFR over 25% from baseline, accompanied by a drop in CKD stage. Furthermore, 48 out of 187 (25.67%) had a sustained increase (measured at least twice, >3 months in between) in eGFR over 25% from baseline, which could be considered as a regression of CKD [30].The total incidence of CKD progression was 25 out of 187 (13.37%).…… The incidence of patients’ prognosis over CKD stage 3 or over is 57 out of 187 (30.48%)”. All revisions are marked in red using the “Track Changes” function in MS word for a clearer view.
Point 5: Results - For pathological diagnosis, 69 (36.90%) were diagnosed of IgA nephropathy……..(lines 140-144), here, the % values looked not correct. IgA nephropathy, 69 out of 130, so the % should be 63.08. Similar changing required in other values in the mentioned lines.
Response 5: Thank you very much for your comments, the 36.90% of IgA nephropathy was calculated by 69 out of 187 (the total included patients). Similar calculations were done on the percentage of other pathological diagnoses. To avoid ambiguity, we have added the denominator in the text in Line 164 as “……69 out of 187 (36.90%) were diagnosed of IgA nephropathy……” using the “Track Changes” function in MS word.
Point 6: Results - the level of significant values mentioned in table 1-3, should not be zero; it may be <0.01, <0.001 and <0.0001 etc.
Response 6: Thank you very much for your comments, we have revised the significant value of 0.000 to <0.001 and 0.0000 to <0.0001 in table 1-3 so as to keep consistent with the format of decimal digits like other significant values inside the table. The revision doesn’t change the significant result of the comparison for each parameter. We have also marked the changes in red using the “Track Changes” function in the text for a clearer view.
Point 7: Results - Absent of supplementary figures that are required for evaluation of the manuscript.
Response 7: Thank you very much for your comments, we are very sorry for the absence of supplementary figures. We have re-uploaded supplementary figures as well as supplementary tables at the end of the manuscript. We have also attached supplementary figures and tables below in case of missing.
Once again, thank you very much for your valuable comments and suggestions.

Round 2
Reviewer 1 Report
The authors addressed all the comments.